# Zoonotic Implications of the Co-Circulation of Clade 2.3.4.4b and 2.3.2.1a H5N1 Avian Influenza Viruses in Nepal in 2023

**DOI:** 10.3390/v17111481

**Published:** 2025-11-06

**Authors:** Pragya Koirala, Manju Maharjan, Sharmila Chapagain, Barun K. Sharma, Tirumala B. K. Settypalli, Charles E. Lamien, William G. Dundon

**Affiliations:** 1Central Veterinary Laboratory, Kathmandu 44600, Nepal; paggya2000@gmail.com (P.K.); barunvet@hotmail.com (B.K.S.); 2National Vaccine Production Laboratory, Kathmandu 44600, Nepal; manjumaharjan2042@gmail.com; 3Food and Agriculture Organization of the United Nations, Lalitpur 44600, Nepal; sharmilakafle2@gmail.com; 4Animal Production and Health Laboratory, Joint FAO/IAEA Centre of Nuclear Techniques in Food and Agriculture, Department of Nuclear Sciences and Applications, International Atomic Energy Agency, Wagramer Strasse 5, P.O. Box 100, A1400 Vienna, Austria; t.b.k.settypalli@iaea.org (T.B.K.S.); c.lamien@iaea.org (C.E.L.)

**Keywords:** avian influenza, Nepal, H5N1, clade 2.3.4.4b, 2.3.2.1a, reassortant

## Abstract

Samples collected from two avian influenza outbreaks in Bagmati Province in central Nepal between January and March 2023 were positive for H5N1. Full genomes were generated for both viruses, which revealed that one of the viruses was very similar to clade 2.3.4.4b H5N1 identified in Bangladesh in 2021/2022. The second virus was a reassortant H5N1 virus consisting of four genes (HA, NA, NP, and M) originating from a clade 2.3.2.1a H5N1 and the remaining four genes (NS, PB1, PB2, and PA) originating from a 2.3.4.4b H5N1. Notably, this second virus had a high identity with 2.3.2.1a clade viruses identified in humans and cats in India in 2024–2025. These are the first full genome sequences of H5N1 avian influenza viruses from Nepal and given the recent human infections by 2.3.2.1a H5N1 viruses in the region, these data will be of interest to both public health and veterinary authorities.

## 1. Introduction

The Goose/Guangdong (Gs/Gd) lineage of highly pathogenic avian influenza (HPAI) subtype H5 that was first detected in China in 1996 has evolved into multiple HA clades and has spread globally, causing devastating outbreaks in domestic and wild birds [1]. Viruses are also now being detected in mammals, highlighting the dynamic and evolutionary nature of these viruses and the importance for each nation affected by HPAI to have a clear understanding of the viruses present in their countries [1].

HPAI subtype H5N1 was first detected in Nepal in January 2009 [2]. Since then, recurrent outbreaks have been documented almost annually, predominantly between the months of January and March. GenBank (https://www.ncbi.nlm.nih.gov/genbank accessed 17 October 2025) and GISAID (www.gisaid.org accessed 17 October 2025) submissions indicate that both 2.3.2.1a and 2.3.2.1c H5N1 viral clades have been the cause of the outbreaks. Each outbreak has been effectively managed through the application of stamping-out measures. These interventions include the immediate culling of infected poultry, disinfection of contaminated premises, enforcement of quarantine protocols, safe disposal of carcasses and other infectious materials, imposition of movement restrictions, and active surveillance both within and outside designated containment zones. Nepal’s HPAI control strategy does not incorporate vaccination; instead, it relies exclusively on non-vaccination-based approaches focused on biosecurity and containment.

In March 2019, a fatal case due to an infection by an H5N1 virus belonging to clade 2.3.2.1a was reported in a 21-year-old male who had direct contact with poultry, emphasizing the zoonotic potential and public risk of HPAI H5N1 viruses in Nepal [3]. As a result, the Ministry of Health and Population, in collaboration with the Ministry of Agriculture and Livestock Development and WHO, strengthened surveillance and aimed to perform comprehensive epidemiological investigations.

A notable resurgence of HPAI H5N1 occurred in the country in 2023, causing several outbreaks in commercial poultry. This study was undertaken to molecularly characterize two viruses identified during these outbreaks.

## 2. Materials and Methods

### 2.1. Outbreak Description

In 2023, outbreaks in eighteen commercial poultry farms across five central districts: Kathmandu, Lalitpur, Bhaktapur, Kavrepalanchok, and Nuwakot were reported (Figure 1). Specifically, affected poultry (chicken and turkey) from farms in Bhaktapur and Kavrepalanchok exhibited sudden and high mortality (15–23%) along with clinical signs consistent with respiratory distress. Carcasses were submitted by farmers to the Central Veterinary Laboratory (CVL), Kathmandu, for diagnosis.

### 2.2. Viral RNA Extraction and Molecular Diagnostics

Total RNA was extracted from a tracheal swab (see sample details in Table 1) using the RNeasy Mini Kit (Qiagen, Hilden Germany), following the manufacturer’s instructions. The extracted RNA was subsequently analyzed for the presence of avian influenza H5 and N1 genes using real-time reverse transcription PCR (RT-qPCR) [4], see Appendix A.

### 2.3. Genome Sequencing

RNA extracted from tracheal swabs (samples A146/079/80 and A10_079_80) recovered during the outbreaks in Bhaktapur and Kavrepalanchok was sent in RLT lysis buffer (Qiagen) to the Animal Production and Health Laboratory, Seibersdorf, Austria, for genome sequencing. (Note: samples A146_079_80 and A10_079_80 were renamed A/chicken/Nepal/A146_079_80/2023 and A/turkey/Nepal/A10_079_80/2023, respectively).

Then, cDNA was synthesized from purified RNA by reverse transcription using the random K-8N primer (5′ GAC CAT CTA GCG ACC TCC AC-NNNNNNNN 3′) at a concentration of 2.5 μM and the SuperScript IV First-Strand Synthesis System (Invitrogen, Waltham, MA, USA) following the manufacturer’s instructions. RNA was removed from RNA/DNA hybrids by treating them RNase H for 20 min at 37 °C. Double-stranded cDNA was then synthesized using Klenow fragment (NEB, Ipswich, MA, USA) with 800 nM of K-8N primer by polymerization at 37 °C for 60 min, followed by inactivation at 75 °C for 10 min. PCR amplification of ds cDNA was performed using a Pfu Ultra II Fusion HS DNA polymerase Kit (Agilent Technologies, Santa Clara, CA, USA) with 400 nM of K-primer (5′ GAC CAT CTA GCG ACC TCC AC 3′). The thermal cycler profile was as follows: initial denaturation of 1 min at 95 °C, followed by 45 cycles of 20 s denaturation at 95 °C, 20 s of annealing at 55 °C, and extension of 3 min at 72 °C, ending with a final extension of 3 min at 72 °C. The amplified ds cDNA was then purified using 1.8X AMPure beads (Beckman Coulter, Brea, CA, USA). The purified ds cDNA was used for library preparation to run on an S5 Next Generation Sequencing system (Ion Torrent, Thermo Fisher). Approximately 50–100 ng of the double-stranded cDNA was enzymatically fragmented to 200 bp length, using Ion shear Plus reagents (Thermo Fisher Scientific, Waltham, MA, USA) with an optimized shearing time of 13 min, followed by purification using 1.8X AMPure XP beads. The purified and fragmented DNA was ligated with barcode adapters to prepare barcoded libraries using an Ion Xpress™ Plus Fragment Library Kit and Ion Xpress barcode adapters (Thermo Fisher Scientific, Waltham, MA, USA) and further purified using 1.2X AMPure XP beads. The purified adapter ligated libraries were size selected on Pippin Prep (Sage Science, Inc., Waltham, MA, USA) and were further purified using 1.5X AMPure XP beads. The purified size-selected libraries were then subjected to amplification by 8 cycles using Platinum™ PCR Super Mix High Fidelity kit and further purified with 1.2X AMPure XP beads. The purified amplified barcoded libraries at a concentration of 100 pM were pooled in equal volumes to load onto the Ion Chef for automated template preparation and chip loading using the Ion 540™ Kit-Chef (Thermo Fisher Scientific, Waltham, MA, USA). With Ion ChefTM, the pooled libraries were clonally amplified on Ion spheres (ISPs) by emulsion PCR followed by automated loading of template-enriched ISPs onto the Ion 540 chip. The sequencing was performed on the S5TM Next Generation Sequencing system with 500 flows to generate 200 bp reads.

The raw sequences were cleaned to remove low-quality reads (Phred < 20), SISPA adapters, and short reads (<50 bp) using fastq-mcf v1.04.676 (ea-utils) and cutadapt v3.5. The quality of the reads was assessed with FastQC (v0.11.5). De novo assemblies were performed using Megahit (v1.2.9) and Spades (v3.15.5). Using the de novo assembly’s contigs, BLAST searches were performed, which identified the complete genome (GISAID: EPI_ISL_19474871) as the appropriate reference. After mapping the cleaned raw reads against the reference sequence using BWA (v0.7.17), SAMtools (v1.11) was used to generate Mpileup files and perform variant calling using BCFtools (v1.9), filtering only variants with a mapping quality >20 and a minimum coverage depth of 100. The consensus sequences, from reads with a mapping quality >20, were produced using vcfutils.pl (VCFtools v0.1.16) and seqtk (v1.3.106) and compared to the de novo assemblies. The mapping quality and coverage were assessed with Qualimap v2.3 [5]. Additionally, the .bam files were loaded into Integrative Genomics Viewer (IGV software version 2.3; The Broad Institute, Cambridge, MA, USA) for read visualization.

### 2.4. Phylogenetic Analysis

Phylogenetic analysis for the HA sequences was performed using the maximum-likelihood (ML) method available in MEGA 11 [6] employing the Tamura-Nei model of nucleotide substitution with uniform rates among sites and 1000 bootstrap replications. HA sequences from other H5NX viruses were retrieved from GenBank and GISAID.

### 2.5. Identification of Mutations

The FluServer tool (https://platform.epicov.org/epi3/cfrontend#252f46, accessed on the 22 October 2025) available in GISAID was used to identify phenotypically or epidemiologically interesting candidate mutations.

## 3. Results and Discussion

Epidemiological investigations identified two primary transmission routes of the viruses into the farms. The predominant route most likely involved indirect transmission via fomites, such as contaminated vehicles, feed, equipment, and human activity, which facilitated the spread of the virus into and between farms. A second potential transmission pathway was through contact with wild birds, which are known natural reservoirs of HPAI viruses. Nepal is a part of the Central Asian Flyway, which overlaps with the East Asian–Australasian Flyway that covers much of eastern Asia.

A complete coding sequence was obtained for the polymerase basic protein 2 (PB2), polymerase basic protein (PA), and the matrix (M) protein for both viruses. For A/turkey/Nepal/A10_079_80/2023, a complete NEP and non-structural protein 1 (NS1) was also obtained, while a complete PB1, HA, and nucleoprotein (NP) was obtained for A/chicken/Nepal/A146_079_80/2023. For the HA, NP, and neuraminidase (NA), of A/turkey/Nepal/A10_079_80/2023, 18, 12 and 30, nucleotides, respectively, were unresolved at the C-terminal of the open reading frame (ORF). For the NA of A/chicken/Nepal/A146_079_80/2023, 3 nucleotides were unresolved at the C-terminal of the ORF, while for the PB1 and NS, 5 and 54 nucleotides were unresolved at the N-terminal of the ORF, respectively.

Phylogenetic analysis using the HA sequence of both viruses demonstrated that A/chicken/Nepal/A146_079_80/2023 belonged to clade 2.3.4.4b and that it was highly similar (99.82% nucleotide identity) to H5N1 viruses identified in Bangladesh in 2021–2022 but also to viruses from China and Russia (Figure 2). The HA of A/turkey/Nepal/A10_079_80/2023 belonged to clade 2.3.2.1a and was also similar to viruses identified in Bangladesh between 2019 and 2025 but, notably, had the highest nucleotide identity (99.99%) with A/Victoria/149/2024, an H5N1 virus which was identified in a 2.5-year-old child returning from Kolkata, India to Melbourne, Australia, in February 2024 [7]. The child presented with severe influenza and respiratory failure requiring mechanical ventilation but made a full recovery. The HA of A/turkey/Nepal/A10_079_80/2023 was also highly similar (99.98%) to two other H5N1 viruses identified in humans in India in 2025 (EPI4264891_A/India/Mangalgiri_NIV_25_594/2025 and EPI4409319_A/India/UdupiNIV/876/2025) and two in Bangladesh (EPI4374846_A/Bangladesh/Khulna/IEDCR-icddr_b-IC1/2025 and EPI4388596_A/Bangladesh/Khulna/IEDCR-icddr_b-IC2/2025) in 2025 in addition to viruses from two infected cats in India, also in 2025 (EPI4645592_A/cat/Chhindwara/ZD25-2/2025 and EPI4645584_A/cat/Chhindwara/ZD25-1/2025) (Figure 2). Both human cases in India were fatal.

BLAST analysis (within the GISAID database) of the nucleotide sequence from each of the genome segments revealed that A/chicken/Nepal/A146_079_80/2023 was very similar at the nucleotide level (99.70 to 99.91%) to clade 2.3.4.4b H5N1 viruses identified in Bangladesh in 2021 to 2022 (Table 2 and Figure 2). In contrast, virus A/turkey/Nepal/A10_079_80/2023 was identified as a reassortant virus, with the HA, NA, NP, and NS related to 2.3.2.1a viruses identified in India and Bangladesh, and the PB1, PB2, PA, and M related to 2.3.4.4b H5N1 viruses from eastern Asia (e.g., Japan, Vietnam, Russia).

Phylogenetic analysis of the remaining seven segments confirmed the findings of the BLAST analysis. A/turkey/Nepal/A10_079_80/2023 was highly similar to reassortant 2.3.2.1a/2.3.4.4b H5N1 viruses identified in humans and cats in 2025 in India and Bangladesh, while the segments of A/chicken/Nepal/A146_079_80/2023 belonged solely to clade 2.3.4.4b (Appendix A).

Each genome was analyzed for the presence of molecular markers associated with increased virulence and host tropism. Using the FluServer tool, notable mutations were identified (Appendix A). (Given the high similarity of A/turkey/Nepal/A10_079_80/2023 to human viruses, A/Victoria/149/2024 was included in the analysis for comparative purposes.) The HA cleavage site for A/chicken/Nepal/A146_079_80/2023 was KRRKRG/LF, similar to highly pathogenic 2.3.4.4.b viruses circulating globally, while for A/turkey/Nepal/A10_079_80/2023 it was RRRKRG/LF, similar to 2.3.2.1a viruses, including A/Victoria/149/2024. Importantly, mutations (i.e., E190D, Q226L, N224K, and G228S) associated with a shift in preference from α2,3-linked receptors to α2,6-linked sialic acid were not present in either virus [8,9]. However, A/turkey/Nepal/A10_079_80/2023 (and A/Victoria/149/2024) had a V226I mutation that has been associated with a small increase in α2,6-linked sialic acid binding [10]. Additionally, a D170N mutation in A/turkey/Nepal/A10_079_80/2023 (and A/Victoria/149/2024) created a new potential N-glycosylation site, which could potentially affect antigenic and other properties of the virus.

The PB2 of A/chicken/Nepal/A146_079_80/2023 and A/turkey/Nepal/A10_079_80/2023 (and A/Victoria/149/2024) contained the avian-specific host markers 271T, 627E, 631M, 701D; in contrast all three viruses had 89V, 309D, 339K, 477G, and 495V substitutions which have been shown to enhance the polymerase activity and increase virulence in mice [11]. In the PA of A/turkey/Nepal/A10_079_80/2023, a K615R change has been linked with a change in the efficiency of the viral polymerase, which can, in turn, affect host specificity and pathogenicity [12]. Again, this mutation was also seen in the human virus A/Victoria/149/2024. A new potential N-glycosylation site was identified in the NA of A/chicken/Nepal/A146_079_80/2023 at position 70 due to a serine to asparagine mutation. Interestingly, this same N-glycosylation site was removed in A/turkey/Nepal/A10_079_80/2023 (and A/Victoria/149/2024), due to an S70P mutation. However, a threonine to an asparagine mutation created a new potential N-glycosylation site at position 361 in the NA of A/turkey/Nepal/A10_079_80/2023 (and A/Victoria/149/2024). In the NP, there was a Y52H mutation observed in both viruses (including A/Victoria/149/2024) that has also been reported in dairy cattle in the US and pinnipeds in Europe and North America, but its significance has yet to be fully elucidated [13]. A/chicken/Nepal/A146_079_80/2023 and A/turkey/Nepal/A10_079_80/2023 (and A/Victoria/149/2024) had a serine (S), a phenylalanine (F) and a methionine (M) at positions 42, 103 and 106, respectively, of the NS1 which has been reported to increase virulence by interacting with antiviral factors that trigger interferon production [11,14].

Despite the notable similarities between A/turkey/Nepal/A10_079_80/2023 and A/Victoria/149/2024, there were still several differences between the viruses, as shown in Appendix A. In total, FluServer identified 30 mutations in A/Victoria/149/2024 and A/turkey/Nepal/A10_079_80/2023 that were not shared by each other. These mutations could be investigated further to determine whether they play a role in host specificity and/or increased pathogenicity.

Since its arrival in North and South America in 2022, clade 2.3.4.4b H5N1 viruses have now become the dominant subtype globally [15]. Recently, clade 2.3.4.4b viruses have been identified in Bangladesh and India [16,17]. The first clade 2.3.4.4b identified by Barman et al. (2023) in Bangladesh in 2021, and now referred to as genotype BD-1, was similar to H5N1 clade 2.3.4.4b viruses from Europe [16,18]. The authors conducted further surveillance and characterization in Bangladesh between January 2022 and November 2023 and identified three additional genotypes of clade 2.3.4.4b viruses (BD-2, BD-3, and BD-4), highlighting the diverse and dynamic nature of viruses in the region [18]. From the BLAST and phylogenetic analysis of A/chicken/Nepal/A146_079_80/2023, it was revealed that this virus belonged to the original genotype BD-1 identified in Bangladesh in 2021 (Appendix A).

In conclusion, the current study has identified both a clade 2.3.4.4b and a clade 2.3.2.1a/2.3.4.4b reassortant virus (Figure 3) in Nepal for the first time. The clade 2.3.2.1a/2.3.4.4b reassortant virus was highly similar to viruses that have caused recent human fatalities in India, and so, these findings should encourage Nepalese authorities to increase surveillance and molecular characterization of AIVs present in their country. The study should also encourage further studies in order to better understand the impact of circulating avian influenza viruses and their zoonotic potential.

## Figures and Tables

**Figure 1 viruses-17-01481-f001:**
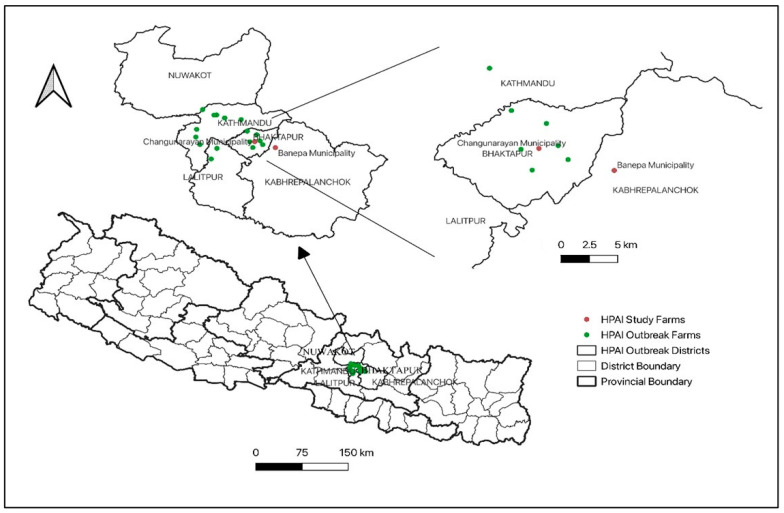
Outbreaks of HPAI in five districts of Nepal in 2023. The red dots indicate the two farms from which the samples were taken for this study.

**Figure 2 viruses-17-01481-f002:**
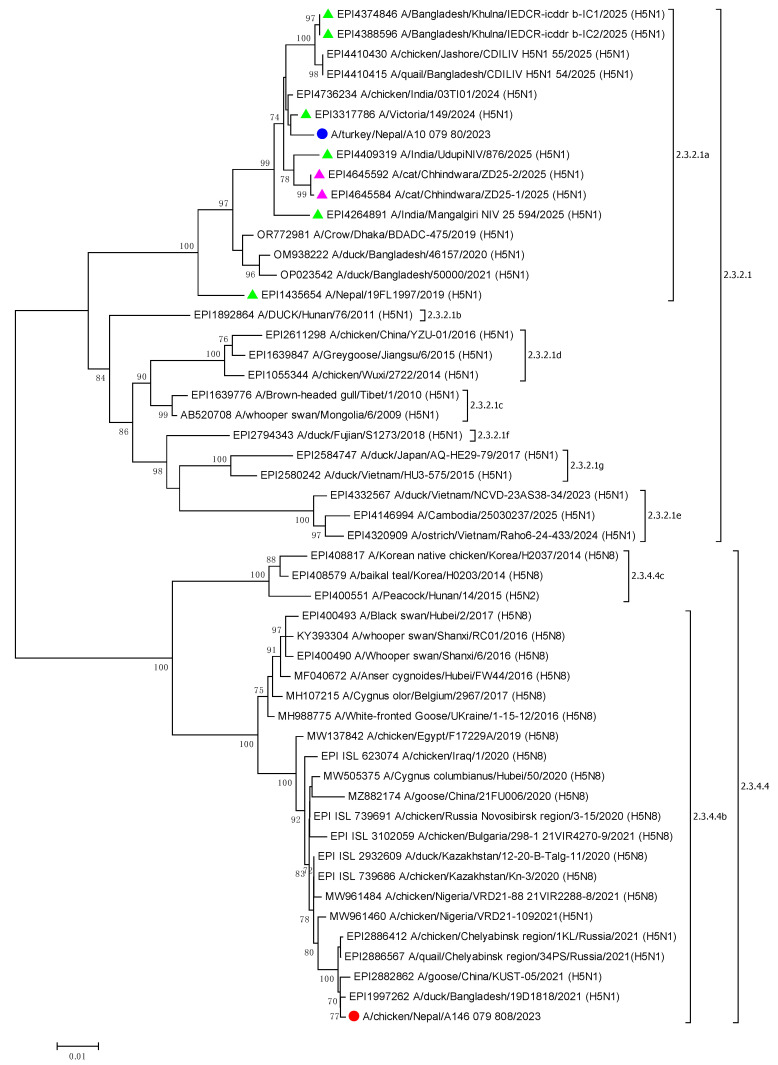
Maximum-likelihood phylogenetic tree of the complete HA gene (1691 bp). The two samples from this study are highlighted with a red and a blue dot. Human and feline samples are shown with green and pink triangles, respectively. Clades are identified, and bootstrap values > 70 are shown.

**Figure 3 viruses-17-01481-f003:**
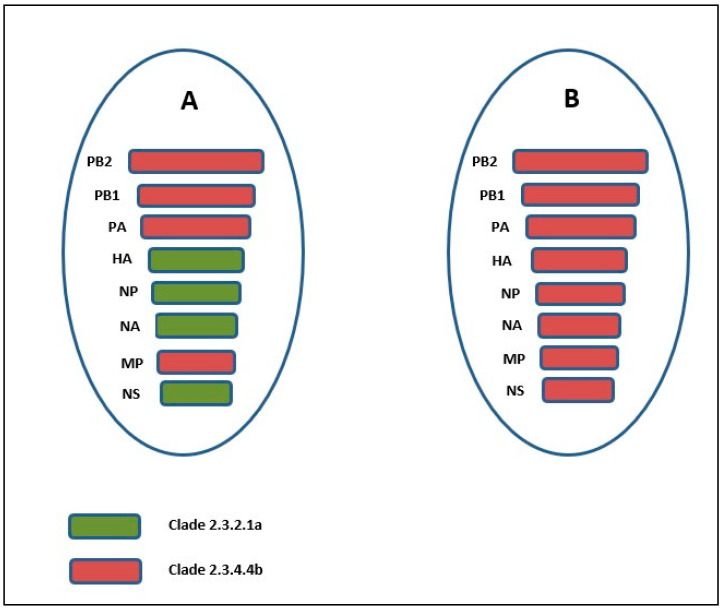
Schematic representation of the origins of each segment of (**A**) A/turkey/Nepal/A10_079_80/2023 and (**B**) A/chicken/Nepal/A146_079_80/2023.

**Table 1 viruses-17-01481-t001:** Susceptible populations, morbidity, and mortality figures from two representative outbreak cases.

Sample Code	Administration Region	Location	No. of Susceptible	No. of Deaths	No. of Culled	Date of Sample Collection	Sample Source	Sample Type
**A146/079/80_H5N1**	Bagmati Province	Bhaktapur, Chagunarayan Municipality	6131	1404	4727	27 January 2023	chicken	Tracheal Swab
**A10/079/80_H5N1**		Kavrepalanchok, Banepa Municipality	4306	676	3630	21 March 2023	turkey	Tracheal Swab

**Table 2 viruses-17-01481-t002:** Comparison of the genome segments of A/turkey/Nepal/A10_079_80/2023 and A/chicken/Nepal/A146_079_80/2023 with their nearest relatives. Clades 2.3.2.1a and 2.3.4.4b are distinguished by green and gray shading, respectively.

	A10/079/80	%	Clade	A146/079/80	%	Clade
PB2	A/duck/Wakayama/22A6T/2022	99.35	2.3.4.4b	A/goose/Bangladesh/19D1819/2021	99.91	2.3.4.4b
PB1	A/chicken/Kagawa/22C2T/2022	99.05	2.3.4.4b	A/duck/Bangladesh/19D1819/2021	99.70	2.3.4.4b
PA	A/pelican/Wakayama/3011003/2022	99.10	2.3.4.4b	A/duck/Bangladesh/19D1819/2021	99.77	2.3.4.4b
HA	A/Victoria/149/2024 A/chicken/India/03TI01/2024	99.29 99.23	2.3.2.1a	A/goose/Bangladesh/19D1819/2021	99.83	2.3.4.4b
NP	A/duck/Bangladesh/O-D-065/2025	99.21	2.3.2.1a	A/goose/Bangladesh/19D1819/2021	99.73	2.3.4.4b
NA	A/chicken/Bangladesh/18-B-569/2022	98.65	2.3.2.1a	A/duck/Bangladesh/51600/2021	99.86	2.3.4.4b
M	A/shoveler/Novosibirsk_region/5029k/Russia/2021 A/duck/Vietnam/HU16-NS82/2023 A/chicken/Wakayama/22B3T/2022	99.40	2.3.4.4b	A/duck/Bangladesh/19D1819/2021	99.80	2.3.4.4b
NS	A/chicken/Bangladesh/O-FM-003/2025	99.07	2.3.2.1a	A/duck/Bangladesh/CE-111-04-DB-11-DU-O/2022	99.87	2.3.4.4b

## Data Availability

The sequences generated in this study have been submitted to GenBank under accession numbers PV423055- PV423062 and PV423064- PV423071.

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
