# Peer review of "Zoonotic Implications of the Co-Circulation of Clade 2.3.4.4b and 2.3.2.1a H5N1 Avian Influenza Viruses in Nepal in 2023"

_viruses, 2025, doi:10.3390/v17111481_

Round 1

Reviewer 1 Report

Comments and Suggestions for Authors

In the article “Evidence for co-circulation and reassortment of H5N1 avian influenza viruses of clades 2.3.4.4b and 2.3.2.1a in Nepal in 2023,” the authors present a study of the recent outbreaks in Nepal.

Here are my comments:

It is not clear from the introduction why you wrote the manuscript, what you did, what your goal is?

  1. Materials and Methods

2.1. Outbreak Description – Give more information. How many outbreaks were there, when and where? Why did you choose only these two farms?

2.3. Genome Sequencing:

Lines 64 – 68:

The code (acronym) of the samples must be written in the same way: A146/079/80 or A10_079_80

The paragraph needs a syntactic review.

  1. Results and Discussion

The section needs a major overhaul.

Lines 121-130 The information provided is more appropriate for Materials and Methods. Why didn't you sequence all the proteins for both viruses?

Lines 147-159 The paragraph needs grammatical and punctuation editing.

164-171 The paragraph is poorly written. More correct, detailed information and discussion of the results are needed.

If you believe your findings are important not only for veterinarians but also for humans, compare the viruses with human isolates, provide more information about mutations or markers that may be related to increased polymerase activity or improved binding affinity, etc.

Author Response

Thank you for your valuable comments. Please find our responses below

It is not clear from the introduction why you wrote the manuscript, what you did, what your goal is?

Response: Thank you for this comment. We have revised the introduction accordingly. See lines 46 to 54

Outbreak Description – Give more information. How many outbreaks were there, when and where? 

Response: we have revised the text to include this information. See lines 57 to 58

Why did you choose only these two farms?

Response: only two farms were chosen due to local logistical issues  

The code (acronym) of the samples must be written in the same way: A146/079/80 or A10_079_80

Response: this has been corrected.

The paragraph needs a syntactic review.

Response:  the paragraph has been revied for syntax.

Results and Discussion

The section needs a major overhaul.

Response: this section has been extensively rewritten due to the new information added. We hope that the grammar and punctuation is correct.

Lines 121-130 The information provided is more appropriate for Materials and Methods. Why didn't you sequence all the proteins for both viruses?

Response: we feel that this belongs to the results section. Unfortunately, due to the quality and quantity of the RNA obtained from the clinical samples, it was not possible to fully resolve the sequence for all of the segments. 

Lines 147-159 The paragraph needs grammatical and punctuation editing.

Response: this section has been extensively rewritten due to the new information added. We hope that the grammar and punctuation is correct.

164-171 The paragraph is poorly written. More correct, detailed information and discussion of the results are needed.

Response: this section has been extensively rewritten due to the new information added. We hope that it is now acceptable.

If you believe your findings are important not only for veterinarians but also for humans, compare the viruses with human isolates, provide more information about mutations or markers that may be related to increased polymerase activity or improved binding affinity, etc.

Response: thank you for this comment. We have now compared the viruses with human viruses and provided more information on mutations in the text and supplementary files.

Reviewer 2 Report

Comments and Suggestions for Authors

This study provides the first full-genome characterization of H5N1 avian influenza viruses in Nepal, documenting the co-circulation of clade 2.3.4.4b and a novel reassortant (2.3.2.1a/2.3.4.4b) strain during 2023 outbreaks. The findings are critical for regional surveillance, given the recent human infections by 2.3.2.1a viruses in neighboring India and Bangladesh. The methodological rigor, phylogenetic analysis, and identification of virulence markers support its contribution to understanding H5N1 evolution in South Asia. The manuscript is well-structured and merits publication after minor revisions to enhance clarity and contextualization. The study’s timeliness and technical soundness justify acceptance after addressing these points:

  1. Virus naming formats vary (e.g., "A146/079/80" vs. "A146_079_80"). Standardize to underscored format throughout, aligning with ICTV/GenBank submissions (PV423055-71).
  2. ‌Expand discussion of molecular markers (e.g., PB2 627E/701D, NS1 42S/103F) by comparing them to known mammalian-adapting mutations in recent H5N1 literature, especially given human cases in the region.
  3. ‌Highlight the 2.3.2.1a/2.3.4.4b reassortant’s implications more prominently, citing Barman et al.’s 2025 Bangladesh study (Reference 18) to emphasize regional evolutionary patterns.
  4. ‌Supplement Table 2 with a schematic or graphical abstract illustrating the reassortment pattern and key mutations (e.g., HA cleavage sites, NA stalk length differences).

Author Response

Thank you for your comments. Please find our responses below.

Virus naming formats vary (e.g., "A146/079/80" vs. "A146_079_80"). Standardize to underscored format throughout, aligning with ICTV/GenBank submissions (PV423055-71).

Response: the formats have been corrected and standardized.

Expand discussion of molecular markers (e.g., PB2 627E/701D, NS1 42S/103F) by comparing them to known mammalian-adapting mutations in recent H5N1 literature, especially given human cases in the region.

Response: thank you for this comment. We have now compared the viruses with human viruses and provide more information on mutations in the text and supplementary files.

‌Highlight the 2.3.2.1a/2.3.4.4b reassortant’s implications more prominently, citing Barman et al.’s 2025 Bangladesh study (Reference 18) to emphasize regional evolutionary patterns.

Response: thank you for this comment. We have added some more text discussing the work of Barman et al (See lines 238 to 248)

Supplement Table 2 with a schematic or graphical abstract illustrating the reassortment pattern and key mutations (e.g., HA cleavage sites, NA stalk length differences).

Response: We have added an illustration indicating the pattern of reassortments (See figure 3) 

Reviewer 3 Report

Comments and Suggestions for Authors

This manuscript by Koirala et al. presents the first complete genome sequences of H5N1 avian influenza viruses from Nepal, revealing the co-circulation and reassortment between clades 2.3.4.4b and 2.3.2.1a. The study is timely, well-structured, and provides valuable insights into the evolution and spread of HPAI H5N1 in South Asia. The molecular and phylogenetic analyses are robust, and the findings have significant implications for both veterinary and public health sectors. The manuscript is suitable for publication in Viruses after minor revisions.

  1. The abstract succinctly summarizes the key findings. However, it would be beneficial to briefly mention the public health implications of the reassortant virus, especially given the recent human cases in the region.
  2. It would be helpful to include a sentence or two about the known zoonotic potential of clade 2.3.2.1a viruses, given the later discussion of human infections.
  3. Please clarify the year in Figure 1 (“202?”) and ensure all sample codes in the text and table are consistent (e.g., A146/079/80 vs. A146_079_80).
  4. Consider adding a brief comparison with the reassortant virus reported by Barman et al. (2025) in Bangladesh to highlight regional trends.
  5. Figure 2 (phylogenetic tree) is essential but not included in the provided text. Ensure it is clearly labeled and accessible in the final version.
  6. Line 66: “RLT Iysis” should be “RLT lysis”.
  7. Line 136: “A146/079/808/2023” appears to have a typo (“808”).
  8. Some references (e.g., Ref 1, 11) are from 2025 – please confirm these are correct or in press.
  9. The analysis of molecular markers related to virulence and host adaptation is a strength of the study. The discussion of NA stalk length and its implications for transmission in mammals vs. poultry is particularly relevant.

The manuscript presents important and novel data on HPAI H5N1 in Nepal. With the minor corrections and suggestions addressed, it will be a valuable contribution to the field.

Author Response

Thank you for your comments. Please find our responses below.

The abstract succinctly summarizes the key findings. However, it would be beneficial to briefly mention the public health implications of the reassortant virus, especially given the recent human cases in the region.

Response: we have revised the abstract based on our new observations and the public health implications

It would be helpful to include a sentence or two about the known zoonotic potential of clade 2.3.2.1a viruses, given the later discussion of human infections.

Response: because of the new observations included in the revised manuscript, 2.3.2.1a viruses that have cause human infections in India, Bangladesh and Nepal have now been highlighted.

Please clarify the year in Figure 1 (“202?”) and ensure all sample codes in the text and table are consistent (e.g., A146/079/80 vs. A146_079_80).

Response: the error has been corrected, and the formats of the viral names have been corrected and standardized

Consider adding a brief comparison with the reassortant virus reported by Barman et al. (2025) in Bangladesh to highlight regional trends.

Response: thank you for this comment. We have added some more text discussing the work of Barman et al (See lines 238 to 248)

Figure 2 (phylogenetic tree) is essential but not included in the provided text. Ensure it is clearly labeled and accessible in the final version.

Response: We are not sure what the reviewer is asking us since Figure 2 (phylogenetic tree) was provided.

Line 66: “RLT Iysis” should be “RLT lysis”.

Response: error has been corrected.

Line 136: “A146/079/808/2023” appears to have a typo (“808”).

Response: error has been corrected.

Some references (e.g., Ref 1, 11) are from 2025 – please confirm these are correct or in press.

Response: All references have been published and are available online.

Round 2

Reviewer 1 Report

Comments and Suggestions for Authors

When you  cite FluServer you should cite the website URL (http://flusurver.bii.a-star.edu.sg).